# Assessment of Phenotype Relevant Amino Acid Residues in TEM-β-Lactamases by Mathematical Modelling and Experimental Approval

**DOI:** 10.3390/microorganisms9081726

**Published:** 2021-08-13

**Authors:** Sara Madzgalla, Helena Duering, Jana C. Hey, Svetlana Neubauer, Karl-Heinz Feller, Ralf Ehricht, Mathias W. Pletz, Oliwia Makarewicz

**Affiliations:** 1Institute for Infectious Disease and Infection Control, Jena University Hospital, Am Klinikum 1, 07747 Jena, Germany; helena.duering@med.uni-jena.de (H.D.); jana.hey@gmx.de (J.C.H.); svetlana_neubauer@hotmail.com (S.N.); mathias.pletz@med.uni-jena.de (M.W.P.); oliwia.makarewicz@med.uni-jena.de (O.M.); 2InfectoGnostics Research Campus, 07743 Jena, Germany; karl-heinz.feller@eah-jena.de (K.-H.F.); Ralf.Ehricht@leibniz-ipht.de (R.E.); 3Instrumental Analysis Group, University of Applied Sciences, 07745 Jena, Germany; 4Institute of Physical Chemistry, Friedrich Schiller University Jena, 07743 Jena, Germany

**Keywords:** antimicrobial resistance, Gram-negative bacteria, molecular diagnostics, extended-spectrum beta-lactamases, beta-lactamase-inhibitor

## Abstract

Single substitutions or combinations of them alter the hydrolytic activity towards specific β-lactam-antibiotics and β-lactamase inhibitors of TEM-β-lactamases. The sequences and phenotypic classification of allelic TEM variants, as provided by the NCBI National Database of Antibiotic Resistant Organisms, does not attribute phenotypes to all variants. Some entries are doubtful as the data assessment differs strongly between the studies or no data on the methodology are provided at all. This complicates mathematical and bioinformatic predictions of phenotypes that rely on the database. The present work aimed to prove the role of specific substitutions on the resistance phenotype of TEM variants in, to our knowledge, the most extensive mutagenesis study. In parallel, the predictive power of extrapolation algorithms was assessed. Most well-known substitutions with direct impact on the phenotype could be reproduced, both mathematically and experimentally. Most discrepancies were found for supportive substitutions, where some resulted in antagonistic effects in contrast to previously described synergism. The mathematical modelling proved to predict the strongest phenotype-relevant substitutions accurately but showed difficulties in identifying less prevalent but still phenotype transforming ones. In general, mutations increasing cephalosporin resistance resulted in increased sensitivity to β-lactamase inhibitors and vice versa. Combining substitutions related to cephalosporin and β-lactamase inhibitor resistance in almost all cases increased BLI susceptibility, indicating the rarity of the combined phenotype.

## 1. Introduction

The β-lactamases (BL) encompasses a large (more than 2800 variants [1]) and phylogenetically highly diverse group of enzymes, which confer antibiotic resistance by hydrolysing the β-lactam ring of β-lactam antibiotics. Within the BLs, the TEM BLs represent one of the clinically oldest groups disseminated worldwide in Gram-negative pathogens [2] with hydrolytic activity toward penicillins, cephalosporins and related antibiotics. The older variants, such as TEM-1, only hydrolyse penicillins and early cephalosporins and are sensitive to β-lactamase-inhibitors (BLI), whereas point mutations can result in an extension of the hydrolytic spectrum towards newer generation cephalosporins, monobactams and inhibitors. Particularly the resistance against inhibitors is of epidemiological and clinical interest. It can be assumed that the TEM variants accompany the currently most frequent BLs of the CTX-M-group in almost 50% of Gram-negative isolates [3]. The CTX-Ms generally confer resistance to certain third generation cephalosporins (e.g., cefotaxime = CTX) but are mostly sensitive to BLI [1]. As combinations of β-lactams with BLI are often used as empiric treatment for Gram-negative infections, BLI resistance mediated by the TEMs is becoming more relevant due to increasingly limiting treatment options.

The phenotype of the TEM variants results from different amino acid substitutions and can be divided into four groups according to the Bush–Jacoby–Medeiros classification: broad spectrum = 2b—hydrolysis of penicillins and early cephalosporins; extended spectrum = 2be—additional hydrolysis of further oxyimino-β-lactams (e.g., cefotaxime, ceftazidime, aztreonam); inhibitor resistant = 2br—resistant to at least one BLI; inhibitor-resistant extended-spectrum = 2ber—extended spectrum and reduced sensitivity to clavulanic acid [4]. The sequences and metadata of allelic TEM variants are collected by the NCBI National Database of Antibiotic Resistant Organisms (NDARO), where currently 192 allelic variants of TEM are listed, of which 134 have an attributed phenotype. 

The NDARO reference database and associated databases (e.g., CARD [5], ResFinder [6], ARG-ANNOT [7]) serve as the basis for resistance prediction algorithms applied on whole genome sequence data (WGS) [8,9]. However, knowledge of a particular gene alone is not enough to predict the phenotypic expression of resistance. The allelic variant and its resistance profile, especially for the complex BL, are essential. However, in the case of TEM variants, the interaction between the individual point mutations makes predictions much more difficult as 63 positions underwent amino acid changes in natural allelic variants. In the past, various phenotypic analyses of natural TEM variants and on mutagenised clones were performed. Thus, most phenotype relevant substitutions in TEM as well as the closely related SHV BL [10] are well known, but the different studies are often difficult to compare to each other, as different models and substrates were used, or only a few substitutions were introduced.

Therefore, the present work aimed to evaluate the role of specific substitutions on the resistance phenotype of TEM variants and to identify new phenotype-relevant amino acid substitutions (PRAS) or to redefine or confirm their function, but also the power of the mathematical prediction models. The PRAS were (i) predicted by mathematical approximation algorithms based on the NDARO reference database and (ii) experimentally confirmed by, to our knowledge, the most extensive site-directed mutagenesis and resistance testing study. 

## 2. Materials and Methods

### 2.1. In Silico Sequence Analysis

From the NCBI National Database of Antibiotic Resistant Organisms (NDARO), 179 amino acid sequences of TEM BL allelic variants were downloaded (https://www.ncbi.nlm.nih.gov/pathogens/antimicrobial-resistance (accessed on 1 April 2019)). The functional classification according to Bush–Jacoby–Medeiros [4] was included for 128 allelic variants in the NCBI data. The complete dataset is provided as Appendix A. 

Sequence alignment was performed by applying the Clustal Omega algorithm for multiple sequence alignment [11]. TEM-1 was set as reference sequence (WP_000027057.1).

A continuous numbering scheme of the amino acid positions based on the alignment, including the signal peptide, was used for mathematical modelling and determination of PRAS. The Ambler scheme [12] was used to evaluate and discuss the results to facilitate traceability and comparison with other related BL classes. 

### 2.2. Mathematical Modelling

The software optiSLang [13] (Dynardo (Dynamic Software and Engineering) GmbH, Weimar, Germany) was used for the prediction of PRAS positions in TEM variants as described previously [10]. Briefly, optiSLang is an application package offering different parametric tools for multi-disciplinary problem resolution that iteratively optimise a model by reducing irrelevant variables by the evolutionary Pareto front algorithm and the Metamodel of Optimal Prognosis (MOP), which can be linked with further interpolation methods (e.g., kriging). A Coefficient of Prognosis (CoP) is calculated after each cycle’s validation providing information on the model quality and robustness.

The input data set (amino acids sequences of the TEMs) was transformed into different cardinal data sets: (i) simply numbered, (ii) by including the isoelectric points (IP) of each amino acid residue, or (iii) the molecular mass (MM) of the amino acid residue. Deletions were treated either with an own value (zero) or excluded (resulting in an incomplete dataset).

### 2.3. Microorganisms and Culture Condition

The clinical *E. coli* isolate ks000079 (Jena University Hospital, Jena, Germany) was used for amplification of the TEM-1 gene, including the natural promoter P_3_. *E. coli* laboratory standard strains ATCC 25922 and ATCC 35218 which were purchased from DSMZ (German Collection of Microorganisms and Cell Cultures GmbH, Braunschweig, Germany) and used as quality controls of MIC testing. The ultra-competent *E. coli* XL1-Blue cells were purchased from Agilent Technologies (Santa Cruz, CA, USA). All strains and their derivatives were stored at −80 °C in 10% glycerin stocks in Mueller–Hinton (MH) broth (Roth GmbH, Karlsruhe, Germany). Cultivation was performed in MH broth or on MH agar plates supplemented with 100 mg/L ampicillin (AMP) (Roth GmbH, Karlsruhe, Germany) at 37 °C overnight (no ampicillin was added for ATCC 25922 and the unmodified XL1-Blue).

### 2.4. Primers and Enzymes

Primers for amplification (Appendix A) were designed using the CLC Main Workbench, version 7.0.2 (Qiagen, Hilden, Germany). Primers for site-directed-mutagenesis were designed using QuikChange Primer Design from Agilent Technologies (https://www.agilent.com/store/primerDesignProgram.jsp). Primers were ordered from Merck (Merck, Darmstadt, Germany) in high-performance liquid chromatography (HPLC)-purified quality in a lyophilised form and were resuspended in pure water to a concentration of 100 µM. 

All restriction enzymes, DreamTaq Green DNA polymerase, thermosensitive alkaline phosphatase (AP), and T4 DNA ligase were purchased from Thermo Fisher Scientific, Inc. (Waltham, MA, USA). Sensiscript Reverse Transcriptase was purchased from Qiagen, and Platinum Taq polymerase was purchased from Invitrogen (Carlsbad, CA, USA). All enzymatic reactions were performed according to the manufacturer’s protocols using the corresponding buffers and supplements provided with the kits.

### 2.5. Cloning and Mutagenesis

The commercially available plasmid pCR™-Blunt II-TOPO (Thermo Fisher Scientific) was used to directly clone the blunt-end PCR product of the TEM-1 as recommended by the manufacturer. After ligation, the TEM-1-bearing plasmids were transformed into Mach1™-T1^R^ *E. coli* cells and positive transformants were selected on AMP-agar plates. From randomly selected colonies, the recombinant plasmids were isolated using the QIAGEN Plasmid Mini Kit (Qiagen) according to the manufacturer’s recommendations and the accuracy of the insertion was confirmed by Sanger sequencing (Macrogen Europe BV). The recombinant plasmid, named TOPO-P3-TEM-1, was transformed into XL1-Blue (Agilent Technologies) *E. coli* cells, resulting in strain SM-1 that was used for the preparation of the variant clone library using the QuikChange II Site Directed Mutagenesis Kit (Agilent Technologies) according to the manufacturer’s protocol. All new clones were transformed into ultra-competent XL1-Blue *E. coli* and grown on AMP-agar plates. The substitutions were confirmed for selected clones after plasmid isolation by Sanger sequencing (see above). 

For the promoter mutants, TEM-1 was amplified with primers SN-tem1-P-for and SN-tem1-stop-T-rev, cloned into pBT plasmid and transformed into XL1-Blue, as previously described [10], resulting in strain SM-2.

### 2.6. Expression Analysis by qPCR

The initial clinical isolate bearing the TEM-1 gene and the confirmed clone SM-1 were used to determine the difference in TEM-1 transcripts. Cells were inoculated at 5 × 10^5^ cfu/mL in 10 mL MH broth and grown at 37 °C under rotation (150 pm) in an orbital shaker. The cells of the clinical isolate were harvested after 6 h and SM-1 cells were harvested after 12 h by centrifugation at 5000 rcf in an Eppendorf centrifuge 5804 R, washed two times in PBS and resolved in TRIzol™ (Invitrogen). Total RNA was prepared using an RNeasy kit (Qiagen) according to the manufacturer’s protocol. RNA quality (RIN value > 7) was assessed by a 2100 Bioanalyzer (Agilent Technologies) and the quantity by using the by an Infinite M200 Pro multimode microplate reader and the NanoQuant Plate™ (Tecan, Männedorf, Swiss). The remaining DNA was digested with DNase I according to the manufacturer’s protocol. The reverse transcription was performed with the primer SN-tem1-stop-T-rev using 50 ng total RNA and the Sensiscript Reverse Transcriptase. The quantitative PCR (qPCR) was performed with a 1 µL aliquot of the reverse transcription reaction with SN-tem1-P-for and SN-tem1-stop-T-rev primers using 1.5 U/µL Platinum™ Taq Polymerase, 250 nm dNTPs and 0.15 x SYBR™ Green (Invitrogen) in a 25-µL total volume in the Rotor-Gene cycler (Qiagen) for 40 cycles, applying 95 °C for denaturation, 55 °C for annealing, and 72 °C for synthesis. The absolute copy number of the transcripts was calculated based on a calibration curve using the serially diluted TOPO-P3-TEM-1, which was previously linearised with *Xba*I (Invitrogen), and the concentration was determined with a Qubit™ dsDNA BR Assay Kit on a Qubit 4 fluorometer (both Invitrogen). The absolute molecule number was assessed by applying Equation (1):(1)$$\frac{\mathrm{molecules}}{\mu L}=\left(\mathrm{Avorgadro}\;\mathrm{constant}\times \mathrm{concentration}\right)/\left(\mathrm{basepairs}\times 660\right)$$

### 2.7. Antimicrobial Susceptibility Testing

The minimal inhibitory concentrations (MICs) of different β-lactam antibiotics were assessed by broth microdilution method as recommended by the European Committee for Antimicrobial Susceptibility Testing (EUCAST). Standard laboratory powders of the antimicrobial substances were purchased and solved as stated in Appendix A. The visual readout of the plates was performed exactly after 20 h incubation. At least two independent experiments were carried out for each mutant on each substrate. If the results varied by one or more dilution steps, a third repeat was performed, and the modus was taken for further calculations with the MIC. 

## 3. Results and Discussion

### 3.1. Overview of the TEM Variants

At the time of analysis (2018), the NCBI database divided the 179 TEM variants into the four phenotypes 2b (*n* = 15), 2br (*n* = 26), 2be (*n* = 78) and 2ber (*n* = 9), of which only 102 were documented in corresponding publications. For 51 variants, the phenotype was unknown. Currently, the database has been expanded by seven new variants of unknown phenotype. Four 2br and two 2be variants that were temporary not listed are now listed again.

In the analysed 178 TEM variants (excluding TEM-1), in total 104 amino acid positions were permutated, of which 41 positions were only permutated in variants with unknown phenotype; thus, no conclusion can be drawn on their impact on the phenotype. Sixty-three positions were permutated in variants with assigned phenotypes, but only a few are strictly associated with a specific phenotype within the characterised variants. Only one position (residue 39) was found to be permutated in all phenotypes and therefore cannot be assumed as phenotype-relevant. (For details, see Appendix A.) Most of the permutating amino acid positions were related to one phenotype (*n* = 43, 68.25%). Within those, eight were found solely in 2b, 24 in 2be, two in 2br, and nine in 2ber variants. These represented the most potent positions for PRAS. Permutating positions found in two (*n* = 13, 20.63%) or three (*n* = 6, 0.95%) phenotypes showed lower frequency showing the following distribution: 5 × 2br/2ber, 4 × 2be/2ber, 3 × 2b/2be, 2 × 2be/2br, 1 × 2b/2ber. Within those present, three excluded 2b, two excluded 2ber and one excluded 2br phenotypes. At these positions, particular amino acid residue might be phenotype relevant. Many of the substitutions occur only in one or a few variants; some of those were of unknown phenotype and are accompanied by other potent PRAS. This makes a reliable prediction based purely on frequency almost impossible. Therefore, mathematical modelling was applied to predict amino acid positions with the highest impact on the phenotype. It considered specific amino acid properties but did not interpret the relevance of a particular residue to a particular phenotype. Therefore, targeted mutagenesis was obligate to assess the impact of specific residues found in variants of the respective phenotype. 

### 3.2. Mathematical Prediction of the PRAS

Different mathematical modelling tools were applied, an automatic approximation by MOP and a targeted inclusion of the kriging regression (K). The validation during the iterative modelling process was set to automatic (MOP chose the best protocol based on the CoP values), or the ‘leave one out’ (LOO) method was selected. The analysis was performed excluding and including additional parameters of the amino acids, such as the IP or the MM of the residues and considering (set as zero) (−Z) or ignoring the missing values of deletions. The resulting CoP values that transfer into the impact factor of a particular amino acid position and its permutation for specific phenotypes are summarised in Figure 1. A threshold for the predictive power was assumed at 0.1, which corresponds to a 10% impact factor. In general, all models yielded similar results with only few variations. Substitutions within the signal peptide were not recognised as being relevant for the phenotype.

For the 2b phenotype, the lowest number of PRAS residues achieved CoP values over the threshold and only when the MM was included. These positions were 164 and 238. However, these were not recognised applying other setups. The CoP values dropped under the threshold if, additionally, the IP was considered, suggesting that these positions are of lower relevance for the 2b phenotype, but the amino acid’s charge and size could play a crucial role at these positions. 

Positions 69, 104, and 164 were the most relevant position independent of the applied modelling, both for the 2be and 2br phenotypes. Position 238 was stronger associated with 2be. When the MM was considered, position 238 but also 165 showed CoP values above the threshold. Position 240 was only weakly allocated to both 2be and 2br phenotypes, while positions 244 and 275 were predominantly more strongly associated with the 2br phenotype. When the IP was considered and the deletions were ignored, position 275 was also slightly related to 2be, indicating that the charge and relative allocation in the protein structure of this residue plays a role. 

Only two positions were strongly related to the combined phenotype 2ber independently of the applied modelling: positions 69 and 164. The positions 238, 240 and 275, with previously predicted lower relevance for the 2be and the 2br phenotypes, also showed weak impact on the 2ber phenotype. Additional weak-impact positions (slightly above the threshold), not related to other phenotypes, were identified for 2ber: 42 and 43, 145, 178, 212, 265 and 284. 

In general, the different OptiSLang modellings described the TEM variants with known phenotype with high accuracy (2be (73 ± 2%), 2br (80 ± 2%) and 2ber (71 ± 6%)), while only 47 ± 1% of the 2b variants were matched consistently in the models (Figure 1). However, overall, the results of the OptiSLang were clearly more accurate compared to the SHV analysis previously performed in the same manner [10], where only 33.1% of the 2b, 49.2% of the 2be and 66.5% of the 2br variants were correctly allocated. The reason might be a higher quality of the TEM variants annotations to a specific phenotype in the NCBI database. For the SHV variants, the annotated phenotype for some variants did not correspond to the published underlying MIC values for the cephalosporins. 

### 3.3. The E. coli Model Systems

For a better comparison of the effects of the mutations within the open reading frame, an *E. coli* model based on the XL-1 Blue and the pCR™-TOPO plasmid was generated. The susceptibility to all antimicrobials tested in this study was previously assessed for the XL-1 Blue strain [10]. The promotor mutations could not be performed in the pCR™-TOPO-plasmid, because the P3-promoter was already present in the cloning vector in front of the zeocine gene. This did not impact the TEM-1 expression but leads to multiple binding sites for the mutagenesis primers, and therefore to multiple PCR amplicons and subsequently to uncontrolled in vivo assembly of these amplicons [14]. Therefore, the pBT-vector was used as previously described [10] to produce the promotor mutant library and the effects of promotor mutations were also assessed in the XL1-Blue *E. coli* background. 

As the antimicrobial susceptibility is generally determined by a purely visual determination of the turbidity in the culture that is interpreted as growth and that strongly depends upon the growth rate, the growth of the model strains bearing the TEM-1 (SM-1 and SM-2) were compared to the clinical *E. coli* isolate ks000079 from which the TEM-1 was derived. Compared to a clinical isolate that reached the stationary phase within less than 12 h, strains SM-1 and SM-2 showed retarded growth with a strongly prolonged lag-phase without reaching the stationary phase until 22 h (Figure 2A). 

Additionally, the expression of the TEM-1 in the SM-1 clone and the clinical isolate were compared by qPCR. Although the total biomass of the model organism grew slower, the expression of TEM-1 gen was 80 times higher in SM-1 than in the clinical isolate when both reached the log-phase (Figure 2B). These suggested that the β-lactam antibiotics can be faster hydrolysed and thus could reach sublethal concentrations before the growth curve of SM-1 reached the stationary phase. To avoid misinterpretation, the manual readout of the MIC values for the clone library was therefore performed after 20 h incubation to ensure that the turbidity indicating growth was sufficiently visible but the effect of the increased hydrolysis of the β-lactams did not lead to plasmid loss and enhanced growth of the background strain XL1-Blue.

The TEM-1 in SM-1 and SM-2 strains resulted in resistance to all penicillins (MIC > 1024 mg/mL) (Appendix A). The MICs of CTX and CRO were only marginally influenced by the TEM-1 presence. Compared to XL-1 Blue, SM-1 showed an 8-fold increased MIC_CAZ_ (2 mg/mL), 16-fold increased MIC_FEB_ (1 mg/mL), 8-fold increased MIC (0.5 mg/mL), and 32-fold higher MIC_BPR_ (2 mg/mL). SM-2 showed only a 4-fold increased MIC_CAZ_ (1 mg/mL) but a similar MIC_FEB_. The MIC_ATM_ was four-fold (0.25 mg/mL) and MIC_BPR_ eight-fold higher (0.5 mg/mL). Although BPR is a fifth-generation cephalosporin, it was hydrolysed with similar efficiency as CAZ by the TEM-1, as also shown by others [15]. Ceftaroline, another fifth-generation cephalosporin, was not included in this study as it was hydrolysed by TEM-1 similarly well to the penicillins, so that the MICs were too high to produce clinically relevant changes due to the mutations [16].

Due to the high expression of the TEM-1 gene in SM-1 and SM-2, the required concentration of BLI to achieve inhibition of the BL activity was expected to be higher. Additionally, both plasmids were supposed to be present in different copy numbers that might also impact the MICs. Therefore, the MIC changes in the checkerboard analysis between β-lactams and BLIs were assessed as isoboles for the SM-1 and SM-2, both bearing the same promoters (Appendix A). Both model *E. coli* strains showed MIC_CLA_ of 32 mg/mL, while the MIC_SUL_ was 512 mg/mL in SM-1 but 128 mg/mL in SM-2, and MIC_TAZ_ was 128 mg/mL in SM-1 but was not achieved in SM-2 until 128 mg/mL. Considering the one-well tolerance, the slightly shifted isobols of SM-1 at higher antibiotic concentrations indicated that the copy number of the pCR-TOPO vector seemed to be only marginally higher. 

As the analysis of the mutants was expressed as a ratio to the parenteral TEM-1 clone (ether SM-1 or SM-2), this variance in the MIC pattern of SM-1 and SM-2 based clone libraries could be neglected. However, the checkerboard analysis showed that the EUCAST dosing scheme with fixed low BLI and varying penicillins concentrations would fail to describe the BLI phenotype in these models. Thus, an individual dosing scheme fixing the penicillins concentrations at a higher level (512 mg/L) and varying the BLIs concentration was preferred. 

### 3.4. The TEM Clone Library

Systematic mutagenesis of the TEM-1 open reading frame was performed based on the OptiSLang results for positions with a CoP value > 0.1. At this point, it is important to understand that an interpolation analysis assigns CoP values at positions with a particularly high influence on a certain phenotype also to other phenotypes independent of the substitutions. To verify the influence of the substitution on the respective phenotypes, experimental evidence is required. Thereby, the amino acids were exchanged with those found in natural TEM alleles with the assigned phenotype according to NDARO at the respective positions (Appendix A). 

Double mutants were generated for selected positions if these combinations occur in natural variants and the corresponding combination had not been investigated before. Unusual double mutants and combinations suspected to result in a different phenotype than the single substitutions on their own were produced. Detailed explanations for each decision are given in the respective substitution sections.

In total, the TEM clone library resulted in 35 single mutants and 13 double mutants (Appendix A). These were investigated for the changes in the substrate specificity against different cephalosporins and penicillins combined with BLIs as relative change in MIC normalised to the respective MIC of TEM-1 for the tested substances. The results were also compared to other studies, where site-directed mutagenesis was performed or naturally occurring single mutation alleles were cloned into a neutral background, and the MIC changes were assessed. Studies solely based on enzymatic activity were unsuitable to be compared to MIC-data.

The results of the extensive substrate analysis of the mutant library (Figure 3 and Appendix A) are presented and discussed in the following sections, each according to the PRAS positions.

A short explanation of the structural relevance (see also Appendix A) and structure–function relationship is included, however excellent reviews have already been published, like those of Salverda et al. [2] or Knox [17].

### 3.5. Mutations with Impact on Resistance to Cephalosporines and Monobactam

#### 3.5.1. Position 164 

The positively charged arginine (R) at position 164 is the most frequently altered amino acid residue in 2be and 2ber TEM variants (Appendix A) and showed high mathematical prediction values for all phenotypes (Figure 1). Thereby, all applied models recognised this position as relevant for phenotypes 2be, 2br and 2ber, while models considering only the mass of the residues assigned this position to the 2b phenotype. Substitutions to structurally and functionally different residues of cysteine (C, sulfhydryl group), histidine (H, imidazole side chain) or serine (S, polar but uncharged) occur at this position. R164 is located at the omega-loop, forming two salt bridges (to E171 and D179), thereby stabilising the loop and the active site [17]. 

Substitution R164S highly increased the MICs of all cephalosporines and ATM, except for BPR (Figure 3A). The exchange to S164 has been shown lead to a collapse of the omega-loop resulting in higher accessibility for larger β-lactam antibiotics [18]. The effect of R164H was slightly weaker but quite similar. The MIC_CRO_ was only twice as high as in TEM-1 and therefore below the threshold, while R164C only increased MIC_CAZ_ but even reduced the MICs of other tested cephalosporins and ATM. Other studies also showed that substitutions at position 164 predominantly increased the resistance to CAZ with weakest impact of R164C [19,20,21,22,23,24]. Thus, the predicted structural variability at this position related to the 2be phenotype was in line with the experimental data. None of the substitutions led to increased resistance towards BLIs (Figure 3B). However, the high CoP value of the mathematical modelling for the 2br phenotype might be interpreted as R164 being decisive for this phenotype. 

#### 3.5.2. Position 104

The mathematical prediction also assigned position 104, which only substitutes to E104K, a high impact on the 2be and 2br phenotypes (Figure 1). Thereby, both mass and charge were related to 2be but only charge to 2br (Figure 1). E104 is in a conserved loop (101–111) [25] and is supposed to be in contact with asparagine (N) 132, which is in the conserved SDN loop (130–132) involved in the substrate binding [26].

The E104K substitution is an exchange from negatively to positively charged residue of a similar size and occurs in 39 of the 2be (50%) but also in two 2ber (22%) TEM variants (Appendix A). In agreement with other studies [20,24,26,27], this substitution significantly increased the MICs of CAZ, CTX, ATM and CRO (Figure 3A), but did not increase the resistance to the BLI. This seemed to conflict with the mathematical predictions. However, the high CoP for 2br might indicate that E104 is crucial at this position for the 2br phenotype.

#### 3.5.3. Position 238

The glycine at position 238 permutates in 30 (39%) of the 2be and in two (22%) of the 2ber TEMs variants as well as in 5 variants with unknown phenotype (Appendix A). The exchanges are primary to serine (35 variants), but single mutants with substitutions arginine, aspartic acid (D) or asparagine (N) are known. G238 is on the inner side of the B3-β-strand. This position was predicted by all models to have the highest effect on the 2be, and to a weaker extent and with fewer models on 2ber phenotypes. Considering the mass, increased CoP values were also determined for 2br and 2b (Figure 1). As the smallest amino acid, glycine in the 2b variants has no specific residue, thus exchanges to any other residues will come with a mass increase that explains the CoP values for the 2b phenotype. This position also plays a key role for the 2be phenotype in SHV-variants that are genetically related to TEM [10,28].

The substitution G238S resulted in strongly increased MIC_CTX_ (32–fold) and MIC_CRO_ (eight-fold) but only weakly increased MIC_CAZ_ (Figure 3). These results agreed with other studies, where the impact of G238S was proven on CTX resistance but also on CAZ and CRO [20,21,23]. Moreover, mutation G238S resulted in strongly increased susceptibility to the BLIs. It has been shown that the exchange to serine results in a repositioning of the loop (238–243), creating extra space for bulky β-lactams [29].

The substitution G238R occurs only in the TEM-178, classified as 2ber; however, this mutation did not increase resistance to cephalosporines or BLIs. The 2be phenotype was even abolished, exchanging the uncharged S-residue to an even more prolonged and positively charged R-residue, confirming that a proper charge and mass are essential at this position to open the active site. 

#### 3.5.4. Position 240

Substitutions at position 240 are present in 28 TEM variants, of which 23 (24%) were characterised as 2be, three (33%) as 2ber and two are so far uncharacterised. This position is at the end of β3 sheet and, depending of the amino acid residue, is suggested to play a role in the interaction with substituents of bulky β-lactams [2]. The substitution to lysine (K) is the most prevalent occurring in 26 of the variants (including the three 2ber and both uncharacterised variants), while substitutions to glycine (G), arginine (R) or valine (V) are only known in single 2be variants. Two of these (E240R/V) are accompanied by the substitution R164S, which has been shown to increase the MICs of all cephalosporines. E240V is additionally accompanied by E104K and M182T, of which only E104K is associated with the 2be phenotype. 

The mathematical modelling assigned relatively weak relevance of position 240 for all phenotypes (Figure 1) but considering different properties. Only the automated MOP calculated CoP values over the threshold for 2be phenotypes (both in 2be and 2ber). For 2br phenotypes (both 2br and 2ber) the charge, and for 2b the mass of the residue, were decisive. 

The mutant bearing the substitution E240K showed a weak two-fold increase in MICs of CAZ, CTX, CRO and ATM (Figure 3A), which was in agreement with other studies [19,20,30]. The resistance of E240K to cephalosporines was shown to be more pronounced in the background of other 2be-associated mutations, like R164S or G238S [2,31]. It has been suggested that E240K might compensate for destabilising effects of R164S or G238S substitutions [2]. 

In agreement with work of Baldwin et al. [32], the substitution E240R increased the MIC_CAZ_ four-fold (Figure 3A). In the present study, additionally the MIC_PIP/TAZ_ increased four-fold (Figure 3C). 

The substitution E240V decreased the MICs for all tested drugs. In the 2be variant TEM-149, the mutation is accompanied by the 2be-associated mutations E104K and R164S, which explains its phenotype.

Substitution E240G resulted in highly increased (eight-fold) MICs of CTX, CRO, FEB and BPR, but an only two-fold increase in MIC_CAZ_ (Figure 3A). This was in contrast to the work of Lenfant et al. [33], where the MIC_CAZ_ was increased four-fold, but not MIC_CTX_ and MIC_ATM_. The resistance profile of the natural 2be variant TEM-207, bearing solely the substitution E240G is not published, but considering the present and former data it can be assumed that this substitution confers a general resistance to cephalosporins even of higher generations, and to the monobactam ATM. 

### 3.6. Mutations with Impact on BLI-Resistance

#### 3.6.1. Position 69 

The methionine (M) at position 69 is the most frequently altered position in inhibitor-resistant TEM variants (Appendix A). This includes 17 of the 2br (65%) and 6 of the 2ber (67%) variants. Only one (1%) 2be variant (TEM-169) and four uncharacterised variants possess a substitution at this position. The substitutions include isoleucine (I), leucine (L) or valine (V), all structurally related residues except for the thioether group in methionine. The function of position 69 is elusive, but substitutions could result in a weakening of the cross-link between the active residues S70 and S130 [34]. The mathematical prediction assigned this position a particularly high impact on the 2ber and 2br phenotypes (Figure 1). Accordingly, mutants with substituted M96 showed increased resistance to BLIs. Substitution M69L resulted in eight-fold increased MICs for AMP/SUL, AMX/CLA and PIP/TAZ, whereas for M69I and M69V only MIC_AMX/CLA_ increased (Figure 3C). These results were congruent with other studies [21,35]. The MIC_AMP/AVI_ was also weakly (twice) increased in M69L, suggesting that particularly M69L is associated with overall BLI resistance. 

All substitutions introduced at position 69 resulted in reduced MIC of most tested cephalosporins and ATM, except for CTX (Figure 3A). Although the mathematical models did not consider any MIC data, a relatively high CoP value for this position in relation to the 2be phenotype was predicted. This might be interpreted as a negative impact of a mutation on the 2be phenotype at this position. The relatively low but still measurable CoP value for position 69, even for 2b, confirmed the generally high importance of this position for the phenotype transformation.

#### 3.6.2. Position 130

Serine in position 130 is substituted in three TEM-variants, of which one is characterised as 2be (S130T) and the other as 2br (S130G) [36] (one unknown bearing S130T). The mode of action of inhibitors is to covalently cross-link the hydrolytic Ser70 to Ser130. This can be prevented by substitutions of S130 [37]. 

Mathematically, only a weak relevance for the 2be phenotype was predicted (Figure 1), but because the 2br TEM-76 variant does not bear any other 2br relevant substitutions, this position was mutagenised, and the effects were evaluated. Interestingly, mutation S130T led to high sensitivity to all tested penicillins (Appendix A), thus the MIC of the BLIs could not be assessed. Both substitutions led to a strong decrease in MICs of cephalosporins, except CTX (Figure 3A). The substitution S130T is accompanied by 2be-associated substitutions E104K and G238S in the TEM-211 variant, explaining its phenotype. Substitution S130G resulted in 16-fold increased resistance to AMP/AVI and 4-fold increased resistance to AMX/CLA and PIP/TAZ (Figure 3C), while the MIC_AMP/SUL_ was decreased. These data are in agreement with previous studies [37,38]. It remains unclear why this position was not recognised as potent PRAS by the modelling. Most likely, the low number of variants permutating at this position and the non-relevant S130T substitution corrupted the algorithms. 

#### 3.6.3. Position 240

Two substitutions at position 240 (E204K and E240R) elevated the resistance to PIP/TAZ. The substitutions E204K also elevated the resistance slightly to AMP/SUL. Interestingly, only E240R mutation showed the increasing effect on both MIC of CAZ and PIP/TAZ, indicating its potential to confer the 2ber phenotype. However, the unique TEM-137 variant bearing this substitution beside the R164S mutation has been classified as 2be but not 2ber. Thus, it seems that the impact of the E240R substitution on the BLI-resistance is weak.

#### 3.6.4. Position 244

Position 244 permutates in six 2br (23%) TEM variants and one 2ber (11%) variant (Appendix A) as well as in two uncharacterised variants. The positively charged arginine is exchanged by structurally different residues of serine, cysteine, glycine, histidine (H) and leucine (L). The relevance predicted by the modelling was independent of charge and mass and was primarily related to the 2br and 2ber phenotypes (Figure 1). The long side chain and charge of R244, which is in the B3 β-strand and edge of the binding site, seem to be essential for activation of a water molecule responsible for the proper coordination of the inhibitors and as a result the S170-S130 cross-linkage [34]. 

All mutants of R244 showed an increased resistance to AMX/CLA, but also increased susceptibility to cephalosporins and the monobactam. Simultaneously, the susceptibility to AMP/SUL and PIP/TAZ increased. For substitution R244S, additionally the resistance to AMP/AVI increased (Figure 3C). 

Although substitutions R244S/H/C were shown to be enzymatically less sensitive to the inhibition by CLA, SUL and TAZ [39], they are more sensitive to combinations AMP/SUL and PIP/TAZ. This results from decreased activity towards the respective penicillins [21,38,39,40,41]. 

Substitutions R244L and R244G seem as single mutations to be solely responsible for the 2br phenotype in the variants in TEM 54 [28] and TEM 79 [11]. Thus far, no details have been published against which penicillin/inhibitor combinations these mutations confer resistance. However, in view of the results of the present study, it can be assumed that position 244 is associated with inhibitor resistance, at least for CLA. 

#### 3.6.5. Position 275

Substitutions at position 275 occur in three 2be (4%), six 2br (23%) and one 2ber (11%), and one uncharacterised TEM variants. The arginine permutates to alanine (A), glutamine (Q) or leucine (L). This position was mathematically assigned the highest relevance for the 2br phenotype, followed by the 2be and 2ber phenotypes with higher CoP values when the charge was considered (Figure 1). The CoP values for the 2b phenotype were below the threshold but associated only with the mass. This position is in the last α-helix H11 of the protein that is distant from the active site is suggested to ensure the proper enzyme activity by electrostatic mechanism [42].

In experiments, all substitutions at this position resulted in elevated MIC_AMX/CLA_ and MIC_AMP/AVI_ (two-fold), while mutations R275L and R275Q also resulted in highly increased MIC_PIP/TAZ_ and elevated MIC_AMP/AVI_ (two-fold) (Figure 3C). The substitutions R275Q and R275L occur as single mutations in the 2br variants TEM-122 and TEM-103, each. However, for TEM-122 the MIC testing was based on a clinical isolate [43] where other mechanisms cannot be excluded and only the AMX/CLA combination was tested. For TEM-103 no further information on the phenotype is available. 

The substitution R275Q also occurs in other 2br TEM variants combined with the strongly 2br-related substitution at position 69 and might support the BLI resistant phenotype. The substitution R275A and R275L only occurs in two 2be or one 2ber (TEM-68) variant, respectively. The inhibitor resistance of TEM-68 is only weak [44]. Although the single mutation R275L confers resistance to BLIs and R275A showed weakly increased MIC_AMX/CLA_, the BLI resistance might not prevail in the background of 2be-relevant PRAS in natural variants. 

#### 3.6.6. Position 276

Mutation N276D occurs in four 2ber (44%) and in five 2br (19%) variants but also in one uncharacterised TEM-Variant (Appendix A). This mutation has been suggested to result in a displacement of the helix H11 by a salt bridge to R244, thereby decreasing the affinity for β-lactams and BLI [42]. In the present study, N276D showed no effect on most cephalosporins and reduced MIC_CRO_ and MIC_ATM_, but increased MIC for AMX/CLA, AMP/SULB and PIP/TAZ (Figure 3A,C) which was mainly in agreement with other studies [27,38,45]. However, none of the mathematical algorithms have assigned relevance to this position for any phenotype. This might be biased by the fact that N276D generally coexists with mutations at position 69 in characterised TEM variants and remained ‘hidden’ for the algorithms. 

### 3.7. Mutations with Additive or Synergistic Effects

#### 3.7.1. Position 165 

Tryptophan (W) at position 165 permutates to cysteine, leucine, and arginine in one 2br and four 2ber TEM variants, and to glycine in one uncharacterised (TEM-190) and one 2be variant (TEM-169). Substitutions at this position are generally accompanied by substitutions at 2br-relevant position 69, but the mathematical modelling predicted position 165 a weak relevance for the 2br phenotype (Figure 1). This position is in the omega loop, but there are no hypotheses for its structural or functional impact on the enzyme. 

All created single substitutions (W165C/G/L/R) did not changed visibly the 2br phenotype (Figure 3C) but particularly reduced the MICs of CRO, FEB, BPR and ATM. Interestingly TEM-169 (M69L-W165G) is classified as 2be, but the resistance profile is unpublished and cannot be examined. Double mutations of positions 165 and 69 were not tested in the present work, but the only two-fold increased MIC_CAZ_ of W165G compared to TEM-1 does not disprove the phenotype of TEM-169. 

#### 3.7.2. Position 182

Position 182 underwent permutation in 31 TEM variants with 30 variants bearing substitution to threonine (T) and one to isoleucine (I), both uncharged but leaking the thioether group of methionine (M). This mutation does not occure solely in known 2be or 2br variants but accompanies other known PRAS. This position is in the helix H8 that is far from the active side. Methionine at this position might be responsible for proper folding of the enzyme, particularly resulting from other substituotions [46]. 

As a single substitution, M182T did not increase the resistance to cephalosporines or BLIs (Figure 3A,C). In natural variants, only one known 2be (TEM-126) combines the substitution M182T with D179E. Both were not predicted as PRASs by the mathematical models, but the double mutation D179E-M182T showed increased MIC for all tested cephalosporins with the highest increase (eight-fold) of MIC_CAZ_. There was no effect on MIC_ATM_ (Figure 3A). These results agreed with previous work [40], where the clinical TEM-126 isolate showed resistance to CAZ. The creation of a single mutation D179E failed, although selection of the transformants was done on CAZ and AMP plates (as a reduced MIC_AMP_ was reported for mutation D179E [47]). Possibly this mutation, which is common in SHV variants, is lethal or unstable in TEM as it is present only in TEM-126 [48]. Thus, it was unclear whether the 2be phenotype was caused solely by D179E or if M182T synergistically increased the MICs, which is known to be a strong stabiliser for other enzyme destabilising mutations [49]. 

Combing M182T with an E104K, as occurring in 17 (22%) 2be and three uncharacterised TEM-variants, had no visible synergistic effects (Figure 3B,D). Combining mutation R164C with M182T (as present in three natural 2be variants, of which one (TEM-236) bears only these two mutations) visibly increased the MICs for all cephalosporins including the fifth generation cephalosporin BPR, and ATM (Figure 3B). This indicated a supporting effect of R164C for the 2be phenotype, but simultaneously reducing the MICs of the BLI in both combinations. Position 182 was not recognised by the mathematical models as relevant, indicating the limitations of approximation models to interpret properly rare substitutional constellations with synergistic effects. 

#### 3.7.3. Position 265

Mutation T265M occurs in 11 2be (14%) and three uncharacterised TEM-variants, but also in one 2b and 2ber variant each. Except for the 2b variant (TEM-110), this substitution accompanies different other phenotype-determining substitutions, making it difficult to assign its relevance. The mathematical prediction indicated weak relevance only for the 2ber phenotype (Figure 2). Position 265 is in the last β-sheet B5; thereby, T165 acts as a kind of spacer between H1 and H11, burring a cavity. Exchange to methionine is suggested to increase the packing density between these helices and to stabilise the enzyme [49]. 

In experiments, the single mutation T265M increased inhibitor resistance for AMP/SULB and PIP/TAZ (Figure 3) but only weakly elevated (two-fold) the MICs of CTX and CRO, while reducing the MIC_CAZ_ (by half). These results were in line with other studies [20,50] but contrary to the results of Mulvey et al. [51], who showed increased MIC of CAZ, FEB, AMX/CLA and PIP/TAZ for the TEM-168-transconjugant, which contains only the T265M mutation and is listed as a 2be variant in the NDARO database. However, as the authors stated, this allelic variant is naturally under the control of a stronger promoter, P4. In the present work, the P4 promoter strongly increased the MICs, particularly for the BLIs of the reference TEM-1 variant (promotor mutants are discussed below).

Combining the substitutions T265M and G238S elevated, compared to G238 alone, the MICs of the cephalosporins and slightly (two-fold) MIC_ATM_ but did not impact the MIC of the BLIs. Thus, it seems that substitution T265M has an additive effect on the cephalosporine resistance. Only one 2ber TEM variant (TEM-68) is known so far to contain this mutation, which is accompanied by other PRAS and additionally the substitution R275L (discussed below). Thus, the calculated 2ber-relevance might be biased by the presence of R275L, but it cannot be excluded that the BLI resistance might be elevated when substitution T265M is present besides other 2br-determined substitutions.

#### 3.7.4. Position 275

The impact of the substitutions R275A/L/Q on the resistance to cephalosporins was marginal. Only MIC_FEB_ was two-fold increased and the MIC_ATM_ two-fold reduced by substitution R275Q. The combined substitutions G238S and R275L resulted only in increased MIC of most cephalosporins (except for CAZ) but no significant increase in BLI-resistance was observed. Compared to the single substitution G238S, in the double substitution G238S-R275L the MICs of CRO, FEB and BPR were increased. Thus, at least an additive effect to the 2be phenotype, was confirmed. R275L (also R275Q) is described as a stabilising mutation, which compensates for other protein destabilising mutations [49,52]. Therefore, it could be proposed that R275L supports the extended spectrum phenotype in the background of other mutations.

### 3.8. Combined Substitutions with Antagonistic Effects

#### 3.8.1. Substitutions at Positions 69 and 164 

Interestingly, combing the most phenotype determining substitution for the 2br and 2be phenotypes at positions 69 and 164, respectively, almost abolished resistance to the BLI (activity even lower than TEM-1, except for AVI), and reduced resistance to the respective cephalosporines and ATM, when compared with the single mutants (Figure 3). This illustrates well that combined substitutions often come at a cost of effectiveness for both substance classes. These combinations are present in six 2ber TEM-variants, but the strongly reduced activity towards the BLIs suggest that, also in the natural variants, the MIC of the BLI might be not high and possibly BLI might be still effective against those variants in a higher dose. 

#### 3.8.2. Substitutions L221M and R244H

The lysine at position 221 is altered to methionine (M) in one 2br variant (TEM-145) being accompanied by the mutation R244H previously shown to be related only with AMX/CLA resistance. For this combination, resistance also to PIP/TAZ and AMP/SUL was reported [53], indicating a supportive effect of L221M. The mathematical modelling did not recognise this position as phenotype-relevant, but to prove the TEM-145 phenotype, this double mutation was generated. Both mutations combined resulted in decreased MICs to all tested substances, including BLI and monobactam (Figure 3B,D) indicating a rather antagonistic effect of the combination. It must be noted that the 2br phenotype of TEM-145 was only determined for the clinical isolate, which carried additionally the TEM-1 [53]; thus, the phenotype overlapping and the classification of TEM-145 to 2br can be questioned. 

#### 3.8.3. Substitution at Position 164 Combined with R244S or N276D

Both positions are related to a specific phenotype: position 164 to 2be and 244 to 2br. The combined substitutions R164S and R244S, as found in the estimated 2ber TEM-121 variant, almost abolished the resistance to the cephalosporines and monobactam, when compared to R164S mutation, but also reduced the resistance against AMX/CLA and AMP/AVI (from four-fold to two-fold higher than TEM-1), when compared to R244S mutation (Figure 3). 

In 2ber variants, substitution N276D is accompanied by R164H/S mutations that were shown to be 2be-related. The combinations of two different substitutions at position 164 (R164H/S) with N276D increased the resistance to CAZ and FEB, but reduced the MICs of CTX, CRO and AMT, when compared to R164H/S, respectively, and abolished the BLI resistance, when compared to N276D. N276D has been described to confer structural protein integrity, when combined with other destabilising mutations [49]. The present results indicate that combination of both positions act antagonistic for most substances, except for CAZ and FEB. However, in natural TEM variants, substitutions at these positions are accompanied by 2br-assossiated mutations at position 69, which might compensate for the antagonistic effect. 

### 3.9. Special Cases

When eliminating all PRAS, 41 TEM variants remain, of which three were still classified as 2be (26 have unknown phenotypes): 

TEM-102 bears only four substitutions, with one (A25V) within the signal peptide that becomes removed during secretion and has no impact on the matured enzyme activity. However, the sequence of the signal peptide can impact the efficiency of secretion and thus the concentration of the protein in the periplasm, which correlates with the MIC. Additionally, TEM-102 bears the substitutions H26R (first amino acid of the matured enzyme), A184V (present in six other variants) and L250V (unique in TEM-102); all were mathematically classified as not relevant. As the positions 85, 184 and 250 were not tested in the present work and there is no reference published for TEM-102, so far the phenotype remains elusive. 

In TEM-157 [54], three substitutions were present: V84I, N100S and A184V, all mathematically classified as not relevant. This variant was experimentally proved as 2be by a transconjugation experiments [54]. As V84I and A184V are the only mutation in one 2b variant TEM-116, the impact of substitution N100S was analysed in this work. However, any visible changes in the MIC of the tested substances (Appendix A) were not found for this exchange. Thus, either the 2be variant TEM-157 and/or the 2b variant TEM-116 were wrongly assigned, or the combination of the three substitutions in TEM-157 synergistically increase resistance to cephalosporines. This however has not been proven by mutagenesis so far.

The suspected 2be variant TEM-164 contains only two and unique mutations, L40V and I279T. Surprisingly, the mathematical modelling did not recognise these positions to be relevant for the 2be phenotype. These two uncommon mutations were created in this study, but the 2be phenotype could not be confirmed, neither for the single mutant bearing substitution L40V nor for the L40V-I279T double mutant (Appendix A). This result contrasts with the work of Ben Achour et al. [55], who showed multiresistance to beta-lactam antibiotics by transformation and transconjugation experiments for the native plasmid bearing only the TEM-164 (experimentally proved). Thus far, the different results of both studies cannot be explained.

Although rated with a CoP > 0.1 by the mathematical modelling, the positions 42, 43, 44, 145, 178 and 212 were not mutagenised in this study. They all occur only in the TEM-178 variant, classified as 2ber and additionally bearing mutations at positions 146 without predicted CoP values, and the mutation G238R, which showed only slightly increased MIC_ATM_. The classification of this TEM variant based on a suspicious phenotyping protocol, where the inoculation was increased 100 time to achieve higher MICs for the BLI [56]. Therefore, the correctness of the assignment to the 2ber can be doubted. As this variant showed increased activity towards CAZ and ATM [56], the co-occurring mutations might synergistically affect the G238R substitution, increasing the resistance to cephalosporines, but there is so far no evidence for BLI-resistance based on the underlying substitutions.

### 3.10. Role of Different Promotors

Three point-mutations within the promoter region of the TEM alleles are known, resulting in increased promotor activities and elevated expression of the corresponding TEM genes [57,58]. The P3 promoter is considered the origin, and the promoters Pa/Pb, P4 and P5 as derivatives of P3. The increased expression of TEM leads to a higher demand for inhibitor molecules to saturate the higher concentration of enzymes resulting in elevated BLI-resistance. In the literature, different, and partly contradictory statements are made about the strength of the promoters. Lartigue et al. [57] described a graduate increase in penicillin/inhibitor MICs with P3 < Pa/Pb < P4 < P5, whereas Zhou et al. [58] found a stronger increase in MIC from P3 to Pa/Pb. Both previous works used almost the same cloning system and an isogenic background; thus, it remains unclear why the results differed that strongly.

In the present study, the MIC_AMX/CLA_ increased four-fold for TEM-1 under the control of the Pa/Pb and P4 promoters, and two-fold for P5 promoter, when compared to P3 promoter. The MIC_PIP/TAZ_ was 64-fold increased for Pa/Pb and P4 promoters and only 32-fold for the P5 promoter, while the MIC_AMP/SUL_ and MIC_AMP/AVI_ were not affected significantly and only increased four-fold for all promoter mutants or for Pa/Pb and P4, respectively. Despite, fixing the beta-lactam concentrations and varying the BLI concentrations, the results for promoter Pa/Pb are in agreement with the study of Zhou et al., and with the Pa/Pb and P4 promotors having a stronger impact than P5 on inhibitor resistance in conflict with the data of Lartigue et al. 

## 4. Conclusions

In accordance with previous studies, positions 164 and 238 and to a lesser extent positions 104 and 240 were related to increased resistance against cephalosporins, while positions 69, 130, 244, 265, 275 and 276 confer resistance to BLIs. The position 179 could only be related to cephalosporin resistance when supported by a stabilising substitution (M182T). Comparing to the SHV-variants that are genetically related to TEM [10,28], permutations affecting the phenotype manifested noticeably more frequently in TEM BLs: Only positions 69, 238 and 240 have similar impact on the phenotype in both BL types. Similarly, more substitutions synergistically interact with other mutations in the TEMs. 

Both BL types have been spreading and diversifying within Gram-negative bacteria for decades, with TEM being the first BL detected shortly after the introduction of penicillin in the civilian sector [59]. Therefore, one could assume that the evolutionary permutation is almost exhausted. Interestingly, specific substitutions at position 240 have the potential to increase resistance to BLIs in both BL types. Only one variant (TEM-191) with E240K as a single mutation is present in the database, but no phenotype is given. As the mutation seems to be stable, further unexpected substitutions and/or their combinations in TEM and SHV BLs leading to phenotype changes cannot be fully excluded in the future.

In general, the present work demonstrates that mutations, which increase cephalosporin resistance, result in increased sensitivity to BLI and vice versa. Moreover, the combination of several 2be- and 2br-related substitutions that are present in 2ber-classified TEMs resulted in a 2be phenotype, but the MIC for the penicillin/inhibitor combinations regularly decreased. TEM enzymes conferring combined resistance to BLI and cephalosporins are possible but seem not to apply to all variants classified as 2ber in NDARO. The 2ber phenotype was first included in the functional classification scheme by Bush and Jacoby in 2010 [4]. The same authors relativised it as early as 2013 with the justification that the classification is based on the phenotypic reaction of the producing organisms and thus does not necessarily correlate with the enzymatic properties [60]. There are possibly only two truthful TEM alleles with a 2ber phenotype carrying the substitutions M69L, R164S and N276D (TEM-158 [61] and TEM-125 [62]), that were proved by a mutagenesis similar to this study [21]. Thus, as proven for this combination, supportive mutations can shift the resistance spectrum back in favour of one or the other phenotype. This illustrates how important a deep understanding of the interactions between substitutions within a BL is for interpreting the phenotype from, for example, genome sequence data. 

Misclassifications in the databases bias mathematical predictions. However, the OptiSLang software, that has been developed to solve various arbitrarily complex problems in product design and development, was able to identify the PRAS with high accuracy, despite the presumed misentries. Considering the masses and IPs of the amino acid residues, the accuracy was even further improved, showing the potential and robustness of mathematical modelling when the data sets become too complex. They can facilitate the selection of new molecular diagnostic targets. However, mathematical modelling still must be experimentally proven, as strong PRAS might be underestimated when their prevalence is low. The predictive value might be improved by including further information such as weighting the positions/substitutions. However, the idea was to prove, how a native database would be interpreted by a correlation algorithm and to experimentally prove the predictive value. In view of the increasing flood of data and publications, such automatic approaches are becoming more and more important to provide the essential information, even to researchers who are not intensively familiar with the subject.

The misclassifications result from non-standardised protocols of phenotyping in the past. It is particularly difficult to deduce which TEM phenotype is present based on MIC tests of clinical isolates, as secondary mechanisms, such as promotor activity, efflux and influx, may interfere [63].

From the clinical perspective, however, the phenotype of the culprit pathogen remains most relevant for the therapeutic success. The manifestation of the 2br but possibly also 2ber phenotype relies on the promoter activity that so far is not included in any biobank. The promoters are usually not studied, and they can change rapidly through horizontal gene transfer, which is known for other BLs in combination with transposons and plasmids. Classical biobanks are the wrong format to store such complex information per se. In the face of increasing threats from resistant Gram-negatives, it may be time to rethink and determine phenotyping not purely on MIC but molecularly on whole genome sequencing or on an RNA basis. Combining both, sequencing of RNA can provide the information whether the resistogram is related to mutations of the resistance gene or increased expression. The latter may be easier to address therapeutically through a combination therapy.

## Figures and Tables

**Figure 1 microorganisms-09-01726-f001:**
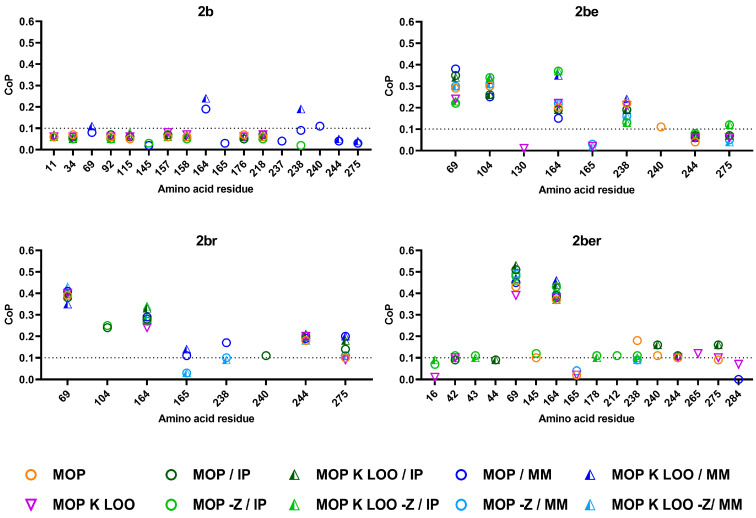
CoP values determined by OptiSLang for the PRAS positions (according to the Ambler nomenclature) in TEM-variants determined by different models. CoP = Coefficient of Prognosis, MOP = Metamodel of Optimal Prognosis, K = kriging regression, LOO = leave one out, −Z = excluding missing amino acid residues (set at zero), IP = including isoelectric point of the amino acid residues, MM = including the molecular mass of the amino acid residues.

**Figure 2 microorganisms-09-01726-f002:**
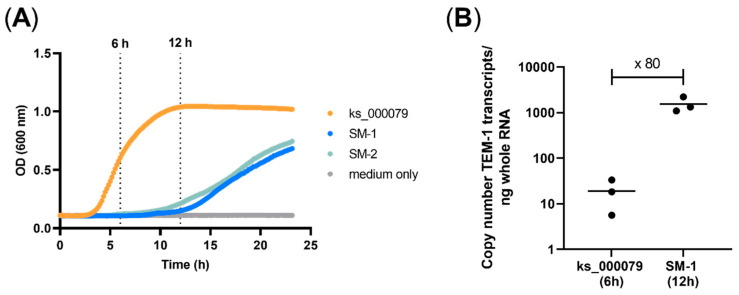
Characteristics of the expression *E. coli* model SM-1 compared to the clinical *E. coli* isolate ks_000079. (**A**) Comparison of the growth curves, (**B**) quantification of TEM-1 transcripts.

**Figure 3 microorganisms-09-01726-f003:**
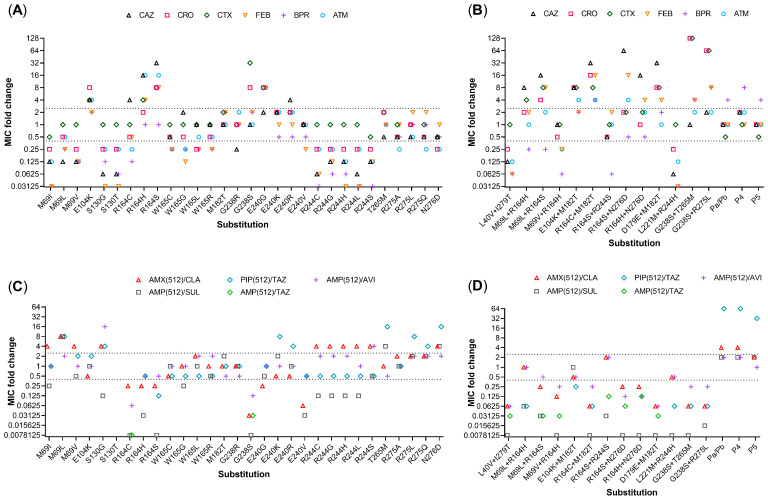
Changes in MIC of single mutants (**A**,**C**) and double mutants (**B**,**D**) compared to the parenteral strains (SM-1, or SM-2 for promotor mutants) against (**A**,**B**) cephalosporins and aztreonam, or for (**C**,**D**) penicillin/BLI combinations. Mutant S130T that was sensitive to all tested penicillins, therefore the penicillin/inhibitor combinations could not be determined. Some mutants were sensitive for PIP, therefore TAZ was combined with AMP.

## Data Availability

All data used in this study are provided with the manuscript and its Appendix A.

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
