# Peer review of "Assessment of Phenotype Relevant Amino Acid Residues in TEM-β-Lactamases by Mathematical Modelling and Experimental Approval"

_microorganisms, 2021, doi:10.3390/microorganisms9081726_

Round 1

Reviewer 1 Report

Madzgalla with co-authors in their work aimed to prove the role of specific substitutions on the phenotype of TEM variants. They also built the mathematical prediction models to identify new phenotype relevant amino acid substitutions.

The undoubted advantage of this work is the huge experimental material on the the most extensive mutagenesis and resistance testing study under uniform experimental conditions ever carried out. Based on the extensive study of MIC's levels in relation to various antibiotics and inhibitors the authors stated that mutations, which increase cephalo sporin resistance, result in increased sensitivity to beta-lactamase inhibitors and vice versa. The authors also showed the dependence of antibiotic/inhibitor MIC's levels on various promoters.

There are some minor inaccuracies and typos in the work:

Line 39 – presently about 2800 beta-lactamases have been isolated and described from clinical bacterial strains

(Bush, K. Past and present perspectives of β-lactamases. Antimicrob. Agents Chemother. 2018, 62, e01076-18.)

Line 63 – there are also another databases concerning beta-lactamases. The LacED database

(http://www.laced.uni-stuttgart.de) contains information collected on the basis of NCBI and PDB on all available TEM and SHV beta-lactamase sequences, including homologous structures and fragments. An ambitious project on summarizing all existing data on all classes of beta-lactamases, which includes biochemical and structural data, is available on the website (http://bldb.eu/). BLDB contains information on more than 2600 unique enzymes, as well as

more than 800 spatial structures belonging to the four molecular classes of beta-lactamases.

It is worth noting the DB (http://www.lahey.org/Studies/), created in 2001 under the leadership of George Jacoby and Karen Bush, for standardization of the nomenclature of the growing number of beta-lactamases, since 2015, this DB has become an integral part of the Bacterial Antimicrobial Resistance Reference Gene Database.

Lines 199-200 – "Only one position (residue 39) was found to be permutated in all phenotypes and therefore cannot be assumed as phenotype relevant.

Some comments:

All TEM type beta-lactamases represent mutants of TEM-1 which differ from it by several single amino acid substitutions. They are divided into two groups: the key mutations and the secondary ones. Key mutations are divided into two types: those (104, 164, 238 and 240), which expand the substrate specificity towards hydrolysis of cephalosporines of III and IV generations, representing so called extended spectrum beta-lactamases (ESBL phenotype 2be)  and the others (69, 130, 244, 275, 276), which lead to the formation of inhibitor-resistant forms (IR phenotype 2br). As a rule, secondary mutations are located distant from the active center and their role is not fully understood. Q39K mutation was the first natural mutation found in beta-lactamases (TEM-2 enzyme). It is the most common among secondary mutations in TEM type beta-lactamases and it was determined in all phenotypes of beta-lactamases. Q39K, M182T, T265M, V84I and A184V do not relay to so called drug resistance mutations providing ESBL/IR phenotype! They are all neutral by itself. It was already known!

In continuation of the above, the authors train their mathematical algorithm on annotated beta-lactamase sequences belonging to different phenotypes. But no distinction is made for key and co-occurring mutations.

Line 571-574. Concerning position 182 see comments above.

Line 652. TEM-102 bears 3 substitution – one L21F in signal peptide and two mature sequence (R164S and T265M). Moreover His26 is the first aa in mature enzyme.

Line 676. Should be TEM-164 not TEM-168.

Line 678-679. "Although rated with a CoP > 0.1 by the mathematical modelling, the position 42, 43, 678 44, 145, 178 and 212 were not mutagenised in this study.". It would be very interesting.

Line 722-724. "position 240 have the potential to increase resistance to BLIs in both BL types. However, no natural 2br or 2ber variants of TEM and SHV carry this substitution as a single mutation so far."

There is! TEM-191.

Reviewer 2 Report

The manuscript describe the influence of several substrates and inhibitors on natural mutants of TEM beta-lactamase. The obtained results of actions of these compounds on many beta-lactamase mutants recieved in common conditions are very interest and useful results. The obtained models for assessment of phenotype are less accurate and it is not clear ability of them to predict phenotype for new mutant.

Additional remarks:

  1. When You describe the mutated residues it is necessary to include description of position of these residues in secondary structures of protein and their location according active site. Otherwise it is difficult to understand the text.
  2. For several mutations it is known the mechanisms which result in changing the substrates and inhibitors interaction with beta-lactamase. It is prefer to to mention them in the text.
  3. Methionine does not have sulfhydryl group.
